# Performance Analysis of LDPC-Coded OFDM in Underwater Wireless Optical Communications

**Jianzhong Guo** [1], **Jinpeng Xiao** [1], **Jing Chen** [2,*], **Xin Shan** [2], **Dejin Kong** [1], **Yan Wu** [1] **and Yong Ai** [2]

1 School of Electronics and Electrical Engineering, Wuhan Textile University, Wuhan 430200, China; jianzg@whu.edu.cn (J.G.)
2 School of Electronic Information, Wuhan University, Wuhan 430072, China
* Correspondence: chen.j@whu.edu.cn

**Abstract:** The performance of Low-Density Parity-Check (LDPC)-coded Orthogonal Frequency Division Multiplexing (OFDM) is investigated over turbulence channels in underwater wireless optical communications (UWOC). The relation between the bit error ratio (BER) and parameters such as the scintillation coefficient, signal-to-noise ratio (SNR), length of LDPC code, and order of OFDM is quantified by simulation. Results show that while the OFDM with subcarrier quadrature amplitude modulation (QAM-OFDM) has slightly better anti-turbulence performance than the OFDM with subcarrier phase shift keying modulation (PSK-OFDM), the LDPC-coded QAM-OFDM has a much better performance than the QAM-OFDM and the LDPC-coded PSK-OFDM, and, at SNR = 12, it decreases the BER by four orders of magnitude compared to the 16QAM-OFDM system when the scintillation coefficient $\sigma_\xi^2 = 0.05$.

**Keywords:** underwater wireless optical communications; OFDM; LDPC; ocean turbulence





## 1. Introduction

Underwater wireless optical communication (UWOC) is a novel communication technology that uses the blue-green light emitted by a light-emitting diode (LED) as a carrier to transmit information [1]. Compared with the traditional communication method using acoustic waves as the carrier, UWOC has the advantages of a large bandwidth, high speed, low latency, strong anti-interference capability, good confidentiality, and low power consumption. With the development of ocean exploration and exploitation, more and more data need to be transmitted, therefore, UWOC has become a research hotspot.

Due to the limitation of the flash frequency of an LED, it is difficult for UWOC to achieve a data rate over 100 Mbps with on-off keying (OOK) modulation at present. In order to improve the throughput, the orthogonal frequency division multiplexing (OFDM) technique was investigated in the UWOC system [2].

When the optical wave propagates through the underwater channel, it is subject to a variety of interferences such as absorption, scattering, and especially the scintillation caused by ocean turbulence. Therefore, forward error-correcting codes are employed to assure the reliability of data transmission. In the literature [3], a low-density parity-check (LDPC) code was studied to mitigate underwater turbulence-induced fading over the generalized-gamma channel. In addition, a review [4] was conducted on the effect of underwater turbulence on optical communications, showing that the exponentiated Weibull (EW) has a rather well fit for irradiance data under all the turbulence conditions.

In this paper, the capability of anti-turbulence is investigated for LDPC-coded OFDM over the EW channel in UWOCs, and the main work consists of two aspects:

(1) A modulation channel model is established to bridge the EW turbulence channel and the received signal for the simulation;

(2)   The bit error ratio (BER) performance is quantified with respect to the scintillation coefficient, signal-to-noise ratio (SNR), length of LDPC, order, and mode of the subcarriers in OFDM.

The rest of this paper is organized as follows. In Section 2, there is a brief description of the block diagram of the UWOC system and the channel model. Simulations are conducted in Section 3, a discussion and some comparisons are made in Section 4, and Section 5 provides the investigation's conclusions.

## 2. System Block Diagram and Channel Model

### 2.1. System Block Diagram

The system block diagram is shown in Figure 1. The binary bit stream from the source is first encoded by the $(n, k)$ LDPC encoder, where $k$ bits are input and $n$ bits are output. Then in the OFDM modulator, the bit stream undergoes a series of operations such as a serial/parallel conversion, subcarrier modulation, and inverse fast Fourier transform (IFFT), and comes out as the signal $s_{OFDM}$. After that, the signal $s_{OFDM}$ drives the LED to emit light; then the light beam passes through the underwater channel and reaches the receiver where the optical signal is converted into an electrical signal $r_{OFDM}$ by a photodetector. During its travel in the seawater, the light beam is distorted by turbulence and polluted by noise. The original bit stream can be restored by the OFDM demodulator and the LDPC decoder. Since the LDPC decoder adopts the belief propagation algorithm in the logarithmic domain, the OFDM decoder should output a log-likelihood ratio for each bit.

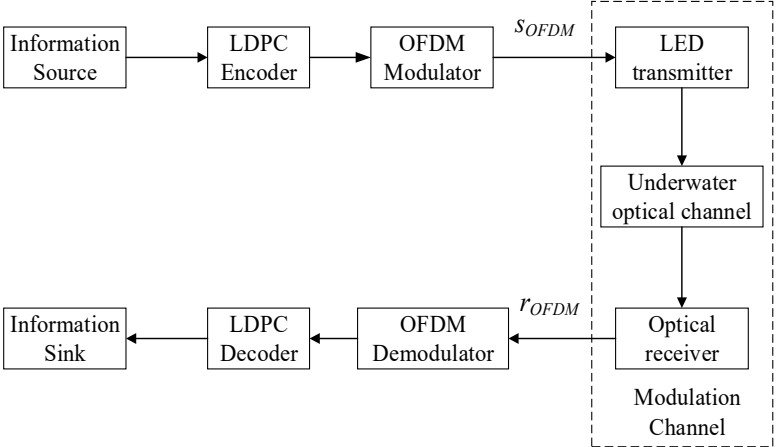

**Figure 1.** Diagram of underwater wireless optical communications.

### 2.2. Underwater Optical Channel

When passing through the seawater, the light beam is subject to scintillating because of underwater turbulence. There are three mathematical models for the distribution of the intensity of the beam under turbulence, the log-Normal model, the Gamma-Gamma model, and the exponential Weibull (EW) model. The log-Normal model is generally used to describe the attenuation channel under weak turbulence [5]. Though the Gamma-Gamma model applies to weak, medium, and strong turbulence [6], it is only effective for point-to-point reception; however, for the aperture-averaged reception, there is a big gap in the experimental data for moderate to strong turbulence. The EW model is a turbulence model proposed in recent years based on the transmission process of laser beams [7]. A large number of experiments and simulations show that the EW model describes the three types of turbulence almost identically to the actual situation when the aperture averaging effect is taken into account.

In the EW model, the probability density of the light intensity $I$ can be represented by the EW distribution shown in Equation (1) [7]:

$$PDF(I) = \frac{\alpha\beta}{\eta}\left(\frac{I}{\eta}\right)^{\beta-1}\left\{1-\exp\left[-\left(\frac{I}{\eta}\right)^{\beta}\right]\right\}^{\alpha-1}\exp\left[-\left(\frac{I}{\eta}\right)^{\beta}\right] \tag{1}$$

where $\alpha > 0$ and $\beta > 0$ are the shape parameters related to the scintillation coefficient, and $\eta > 0$ is the scale parameter related to the mean value of the irradiance. The values of the three parameters $\alpha$, $\beta$, and $\eta$ are closely related to the light intensity scintillation coefficient $\sigma_I^2$, and can be obtained by fitting experimental or simulation data as follows.

$$\alpha \approx \frac{7.220\sigma_I^{2/3}}{\Gamma(2.487\sigma_I^{2/6}-0.104)}$$
$$\beta \approx 1.012(\alpha\sigma_I^2)^{-13/25} + 0.142$$
$$\eta \approx \frac{1}{\alpha\Gamma(1+1/\beta)g_1(\alpha,\beta)}$$

$$g_n(\alpha,\beta) = \sum_{i=0}^{\infty}\frac{(-1)^i\Gamma(\alpha)}{i!(i+1)^{1+n/\beta}\Gamma(\alpha-i)}$$

where $\Gamma(\ )$ represents the Gamma function.

### 2.3. Modulation Channel

In order to investigate the anti-interference capability of the LDPC-coded OFDM system over the underwater optical channel, it is necessary to focus on such a modulation channel as the dash line box in Figure 1, with the input $s_{OFDM}$ and output $r_{OFDM}$. When signals pass through this channel, they will undergo processes of electro-optical conversion, underwater disturbance, photoelectric conversion, thermal noise, etc. The thermal noise generated by the photodetector is modeled as additive Gaussian noise, and the interference caused by other processes (such as light-intensity scintillation) is modeled as multiplicative interference. Therefore, the relation between channel output $r_{OFDM}$ and input $s_{OFDM}$ can be represented as Equation (2), and the details can be found in Appendix A.

$$r_{OFDM} = \xi s_{OFDM} + n \tag{2}$$

where $n$ is the additive Gaussian noise and $\xi$ is the multiplicative interference attenuation coefficient, a random process obeying the EW distribution in Equation (1) in which the scintillation coefficient $\sigma_\xi^2$ is mainly determined by the light intensity scintillation coefficient $\sigma_I^2$ and the photoelectric conversion quantum efficiency.

In the simulation, the SNR is calculated as follows.

$$SNR = \frac{P_{\xi s_{OFDM}}}{P_{\xi s_{OFDM}} + P_n} \tag{3}$$

where $P_{\xi s_{OFDM}}$ is the power of the signal $\xi s_{OFDM}$, a component containing the original signal $s_{OFDM}$ and multiplicative interference $\xi$, and $P_n$ is the power of the additive Gaussian noise.

## 3. Performance Analysis and Simulation

### 3.1. Performance Evaluation and Parameter Setting

In this section, we will evaluate the capability of anti-interference of LDPC-coded OFDM over an underwater channel modeled by Equations (1) and (2). To this end, the relation between the BER and a variety of parameters, as described in Equation (4), will be quantified by simulation according to Equation (2). These parameters involve the channel

(SNR and scintillation coefficient $\sigma_\xi^2$), the LDPC code (the code length $L$ and rate $R$), and the OFDM modulation (the order $Q$, the mode of subcarriers).

$$BER = f(\sigma_\xi^2, SNR, L, R, Q, mode) \tag{4}$$

In the simulation, the BER is calculated by the ratio of error bits and total bits; the SNR is calculated by Equation (3); the LDPC code length $L$ = 256, 512, and 1024; and the rate $R$ = 1/2. The LDPC decoder adopts the belief propagation algorithm in the logarithmic domain, and the maximum number of iterations is 30. In the OFDM, the number of subcarriers is 64, and each subcarrier adopts the same modulation mode and order $Q$, such as 4-phase shift keying modulation (4PSK), 16PSK, or 16-quadrature amplitude modulation (16QAM). The length of the LDPC and order of OFDM is chosen such that they can be best coupled. The simulation results are shown in the following figures.

### 3.2. Simulation Results

The influence of the LDPC code length upon the BER performance is shown in Figures 2 and 3, in which the horizontal axis represents the SNR and the vertical axis is the BER, with scintillation coefficients $\sigma_\xi^2$ = 0.1 and 0.05, respectively. Both figures demonstrate that code length is an important factor that influences the BER performance, for there is an obvious decrease in BER when the code length rises from 256 to 1024.

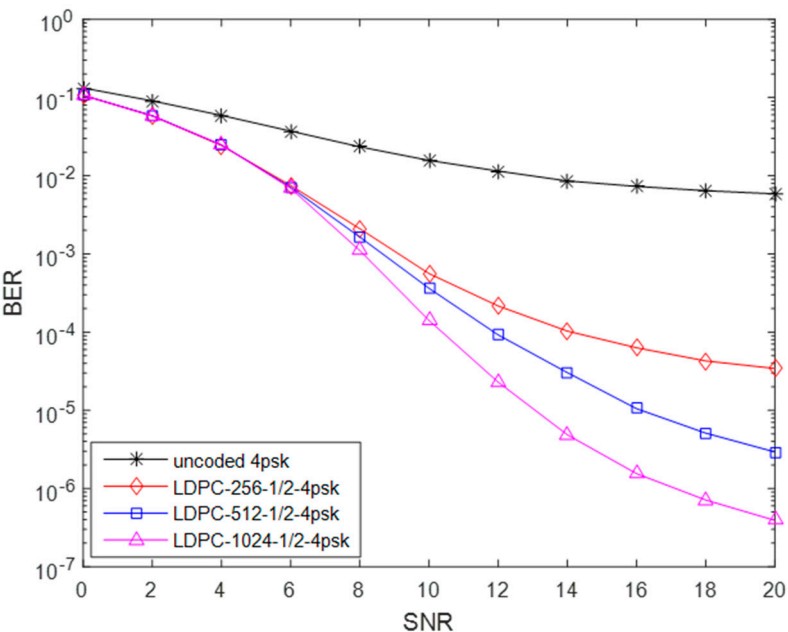

**Figure 2.** BER Performance curve with the LDPC at $\sigma_\xi^2$ = 0.1.

Figures 2 and 3 also show that the scintillation coefficient is the most important parameter that influences the BER performance of the UWOC system. When $\sigma_\xi^2$ = 0.1, the BER goes down very slowly with the increase in SNR, as shown in Figure 2. However, at $\sigma_\xi^2$ = 0.05, the BER goes down quickly with the increase in SNR, as shown in Figure 3.

Figures 4 and 5 depict the influence of the modulation mode and order upon the BER performance of the LDPC-coded OFDM system, where the scintillation coefficient $\sigma_\xi^2$ = 0.1 and 0.05, respectively, and the length of the LDPC is 1024. The figures show that the modulation order has a great influence on the anti-interference capability of the UWOC system. The higher the order, the worse the anti-interference capability is. At the scintillation coefficient $\sigma_\xi^2$ = 0.05, the 4PSK system has such good performance that the BER goes below $10^{-6}$ even without error correction codes and can achieve reliable communication so long as the SNR is large enough. From the figures, it can be seen that the QAM-OFDM has better performance than the PSK-OFDM. There is an interesting phenomenon where

even though the BER performance of the 16QAM-OFDM is slightly better than that of the 16PSK-OFDM, the LDPC-coded 16QAM-OFDM has a BER performance much better than the LDPC-coded 16PSK-OFDM, and this will be interpreted in the subsequent section of discussion.

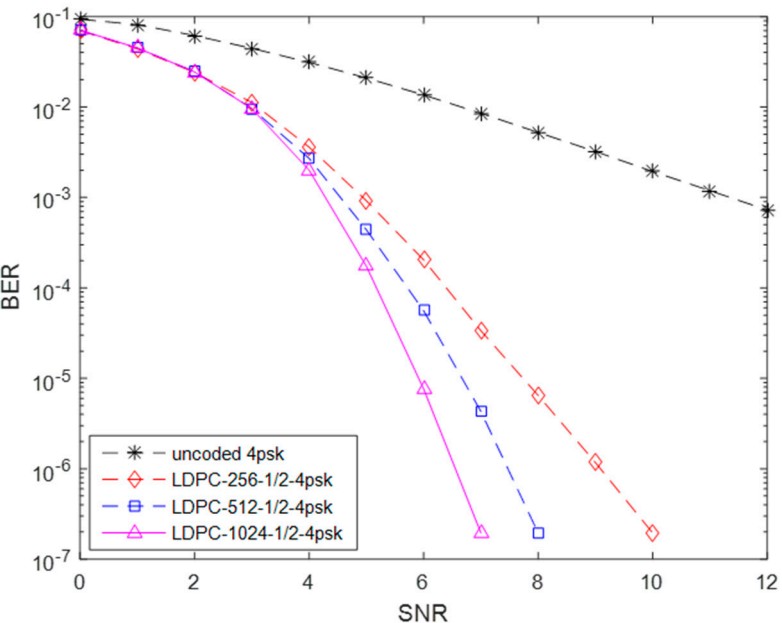

**Figure 3.** BER Performance curve with the LDPC at $\sigma_{\xi}^2 = 0.05$.

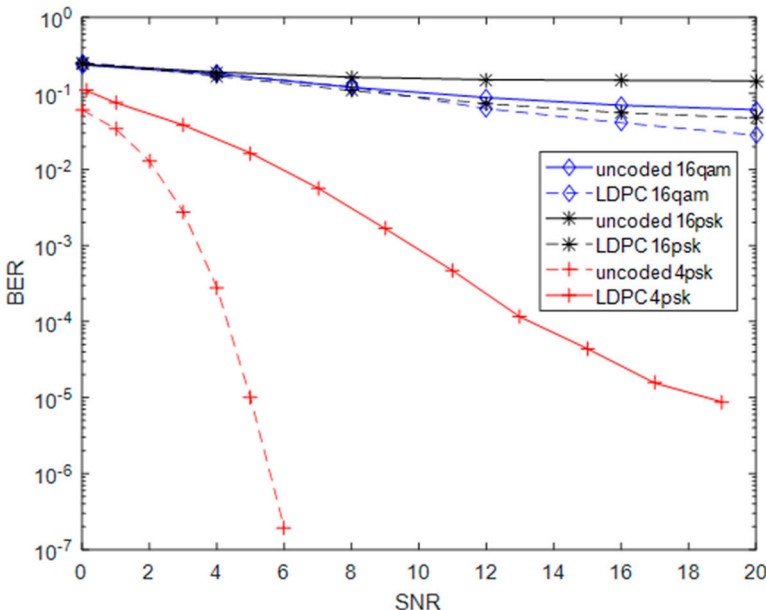

**Figure 4.** BER performance curve with the OFDM at $\sigma_{\xi}^2 = 0.1$.

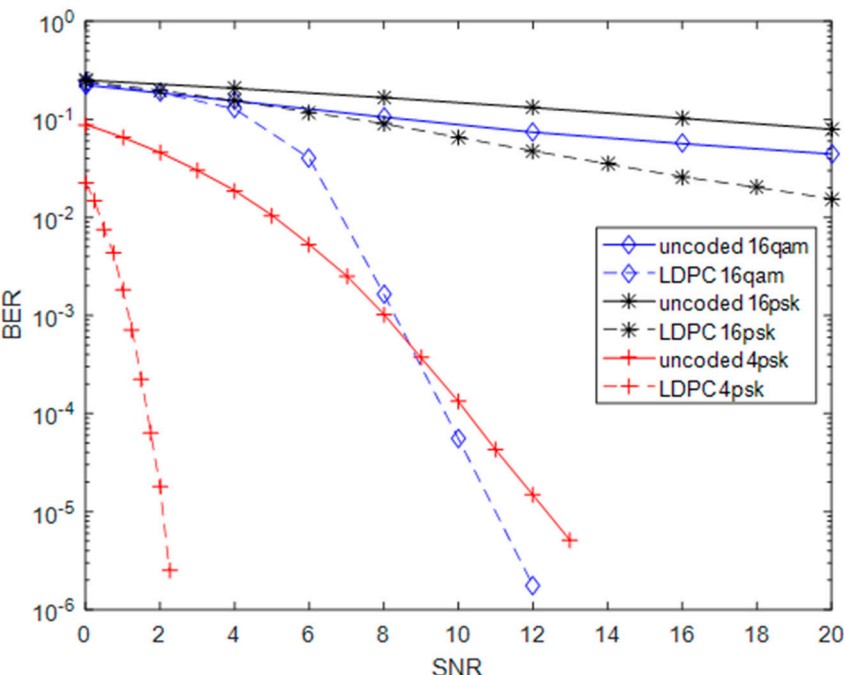

**Figure 5.** BER performance curve with the OFDM at $\sigma_\xi^2 = 0.05$.

Figures 4 and 5 also demonstrate that the order of the subcarrier has a great influence on the anti-interference capability of the UWOC system. The higher the modulation order is, the worse the anti-interference capability is. Obviously, the 4PSK system has a better anti-interference performance than the 16PSK and 16QAM systems. At the scintillation coefficient $\sigma_\xi^2 = 0.05$, the 4PSK system has such good performance that the BER goes below $10^{-6}$, even without error correction codes, and can achieve reliable communication so long as the SNR is large enough.

The two figures also show that the scintillation coefficient is a crucial factor that affects the performance of the system. At the scintillation coefficient $\sigma_\xi^2 = 0.1$ (as shown in Figure 4), the OFDM system alone without the LDPC code cannot achieve reliable communication when SNR < 20dB; the BER goes down so slowly as the SNR increases that it can hardly reach below $10^{-6}$. However, under this condition, even the LDPC-coded OFDM system of 16PSK or 16QAM subcarriers is still unable to reduce the BER to $10^{-6}$ or less, only the LDPC-coded 4PSK-OFDM system can do that.

As the scintillation coefficient decreases to $\sigma_\xi^2 = 0.05$ (as shown in Figure 5), the BER decreases accordingly, but it is still far above $10^{-6}$ for the OFDM system of subcarriers, 16PSK and 16QAM, and cannot achieve reliable communication. However, the LDPC-coded 16QAM-OFDM system can achieve reliable communication; the BER curve goes down quickly like a waterfall. At SNR = 12, especially, the LDPC-coded 16QAM-OFDM system decreases the BER by four orders of magnitude over the 16QAM-OFDM system.

### 3.3. Comparison and Analysis

Equation (4) and the simulation results show that the performance is affected by a series of factors, such as the scintillation coefficient $\sigma_\xi^2$, SNR, the code length of LDPC, order, and mode of OFDM. The scintillation coefficient is a crucial factor; only when it is lower than a certain value can the BER decrease with the increase in SNR. The modulation order has positive and negative effects on the performance of the system: on one hand, a higher order modulation can improve the throughput to overcome the disadvantage of the limitation of the flash frequency of the LED; on the other hand, it decreases the capability of anti-interference of the system. As can be seen from the figures, the higher the order, the higher the throughput is, but the worse the BER performance is. LDPC can improve the

anti-interference capability of the system, and thus compensate for the disadvantage caused by the high order of OFDM, but the degree of improvement is decided by a variety of factors such as the modulation mode and order of the subcarriers, as well as the scintillation coefficient and SNR.

## 4. Discussion

There are a variety of factors that influence the performance of the system, such as the scintillation coefficient $\sigma_\xi^2$, order and mode of OFDM, and length of LDPC. All these factors work together to determine a threshold, and if we design a system with parameters above the threshold, the BER will decrease steeply with the increase in SNR, otherwise, the BER will decrease slowly. This threshold can explain why the LDPC coded 16QAM-OFDM is greatly superior to the LDPC coded 16PSK-OFDM, while the 16QAM-OFDM is slightly better than the 16PSK-OFDM, because the LDPC coded 16QAM-OFDM is above the threshold while the others are not.

Simulation results show that QAM-OFDM is preferable to PSK-OFDM, but which order should be adopted? The 16QAM, 64QAM, or 256QAM? As mentioned above, the higher the order is, the worse the BER performance will be. Therefore, the modulation order should be decided comprehensively. If the higher order is adopted in OFDM, the longer code length should be adopted for the LDPC code to improve the error-correcting ability if the scintillation coefficient and SNR remain the same. However, if the turbulence is strong and the scintillation coefficient is too large, reliable communication cannot be achieved no matter what kind of coding and modulation scheme is used. Under these conditions, adaptive optics technology should be adopted to suppress the effect of turbulence.

## 5. Conclusions

The performance simulation of an LDPC-coded OFDM system is conducted over an underwater optical channel where the intensity of light is assumed to obey the EW distribution. Results show that both modulation and coding have a great influence on the performance of the UWOC system. Though the increase in the modulation order and number of subcarriers can increase the data rate, the capability of anti-interference decreases.

The simulation results of LDPC-coded OFDM modulation show an improvement in the system performance to a certain extent, but it should be noted that the LDPC code can only work under the condition of a small scintillation coefficient. When the scintillation coefficient is higher than this threshold, the LDPC code shows almost no error correction function. Only when the scintillation coefficient is lower than the threshold can the BER decrease with the increase in SNR. For a UWOC system, if the receiving aperture is too small and the transmission distance is too long, resulting in too large of a scintillation coefficient (multiplicative interference), or if the optical signal at the receiver is too weak and the detector is not sensitive enough, resulting in too low a signal-to-noise ratio, reliable communication cannot be achieved no matter what kind of coding and modulation scheme is used.

**Author Contributions:** Conceptualization, J.G., J.C. and Y.A.; Methodology, J.C., X.S. and Y.A.; Software, J.G. and X.S.; Validation, J.G., J.C., X.S. and D.K.; Formal analysis, J.C.; Investigation, J.G. and X.S.; Writing—original draft, J.X.; Writing—review & editing, J.G. and Y.W.; Supervision, J.C. and Y.A.; Project administration, J.C.; Funding acquisition, J.C. and D.K. All authors have read and agreed to the published version of the manuscript.

**Funding:** National Natural Science Foundation of China: U2141255 and 62001333.

**Institutional Review Board Statement:** Not applicable.

**Informed Consent Statement:** Not applicable.

**Data Availability Statement:** Not applicable.

**Conflicts of Interest:** The authors declare no conflict of interest.

**Appendix A**

Suppose the light intensity from the source is *I* and that it follows the EW distribution in Equation (1) when it passes through the underwater turbulence channel. Since the light intensity is modulated by the OFDM signal $s_{OFDM}$ at the transmitter, at the photodetector, the light intensity can be expressed as $Is_{OFDM}$, in which either $Is_{OFDM}$ or *I* follows the EW distribution in Equation (1).

The output of the photodetector can be expressed as

$$y = f(Is_{OFDM}) + n$$

where *n* is the thermal noise caused by the device of the system.

In practice, the output of the photodetector is proportional to the input light intensity under normal conditions. Let the proportional constant be *k*, and we obtain

$$y = kIs_{OFDM} + n$$

$$= \xi s_{OFDM} + n$$

where $\xi = kI$ is stochastic and follows EW distribution.

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
