# Peer review of "Performance Analysis of LDPC-Coded OFDM in Underwater Wireless Optical Communications"

_photonics, doi:10.3390/photonics10030330_

Round 1
Reviewer 1 Report
1) Authors have used an oversimplified channel model. They use Equation (1) to characterize the underwater channel but then employ an extremely simple model that they use in simulations. Starting from Equation (1) how they arrive at Equation (2) needs to be explained further.
2) The literature survey is insufficient. Authors should do a more thorough literature review and include more papers preferably recent ones.
3) In Figure 2, the performance of 16psk, with and without LDPC, and the performance of 16qam, with and without LDPC are quite similar in the sense that there isn't any appreciable improvement in BER even after employing LDPC codes. However, in Figure 3, the performance of 16qam with LDPC is significantly better than that of 16psk with LDPC. This is unusual and needs to be explained.
4) Authors have simulated only one type of LDPC code. Similarly, they have only used 64 subcarriers for the simulations. Given the fact that there isn't much theoretical work in the paper, more simulations with different flavors of LDPC codes and with varying numbers of subcarriers should be included. It would be particularly interesting to see the variation in BER as a function of number of subcarriers.
Reviewer 2 Report
In this work, authors have investigated the performance of LPDC OFDM code with the optical wireless communication under the water. The overall structure of the manuscript is okay. Please check my following comments:
1) Since LPDC codes are not new one. They are widely used with the wireless communication ( Microwave) by many researchers before. Even for the underwater acoustic model, LPDC codes are used long time ago. So my main concern is about the novelty and significance of the current work. The author should highlight the novelty and significance in the Introduction section. In the current manuscript, Introduction is very weak. The author should mention the key works on LPCD codes.
2) The authors should compare the results with the previous works by providing tabular comparison to justify the contributions. At present, I didnt find any new contributions in this work related to the literature.
3) How about the performance of other codes with LPCD codes by using the same simulation setup ?
Reviewer 3 Report
- Revise the abstract focusing on the generic concept of LDPC coded OFDM based UWOC. Compare the performance improvement results between QAM & PSK modulation schemes in terms of numeric values. Elaborate the term LDPC in abstract section.
- Comparison analysis with other related works is missing. Point out the key contribution of this article over the other existing works at the end of introduction section.
- Briefly illustrate the methodology of the system model presented in Figure 1 particularly focusing on LDPC encoder and decoder functionality; otherwise, the block diagram is identical for other encoding schemes in UOWC. Show the mathematical relationship between different blocks used in the diagram. Explain equation 2. Demonstrate the expression of SINR & BER. What is the simulation complexity and power penalty of LDPC coded OFDM based UOWC system? Compare the overall performance of the OFDM based UOWC system with other encoding scheme for benchmarking.
Round 2
Reviewer 2 Report
The authors have revised the manuscript as per the comments.
Author Response
The manuscript has been revised as the reviewer's suggestions.
Reviewer 3 Report
The performance comparison in terms of percentage value between modulation schemes is not yet addressed in abstract section. Related works is still missing which is required to highlight the significance of current research.
